# The Effects of Curing Temperature on CH-Based Fly Ash Composites

**DOI:** 10.3390/ma16072645

**Published:** 2023-03-27

**Authors:** Xiangnan Ji, Koji Takasu, Hiroki Suyama, Hidehiro Koyamada

**Affiliations:** Department of Architecture, Faculty of Environmental Engineering, The University of Kitakyushu, Kitakyushu 808-0135, Japan

**Keywords:** curing temperature, fly ash, cement paste, compressive strength, C-S-H

## Abstract

Curing temperature affects the compressive strength of cement paste systems via the pozzolanic reaction. However, different processes, climates, and weather conditions often result in different initial curing temperatures. The relationship between curing temperature and compressive strength is still an underexplored domain. To explore the effect of curing temperature on calcium hydroxide (CH)-based fly ash composites, fly ashes from different carbon sources were used to make CH-based composites, and the compressive strength, reaction rate, CH content, and C-S-H generation were analyzed. The correlation between the reaction rate and C-S-H content was analyzed. High-temperature curing improved the compressive strength of the cement paste system by affecting the CH-based reaction rate in the initial stage, with the highest initial reaction rate reaching 28.29%. However, after cooling to constant temperature, high-temperature curing leads to a decrease in CH and C-S-H content. The average decrease rate of calcium hydroxide content under high temperature curing is 38%, which is about 2.38 times that of room-temperature curing conditions. This led to a decrease in the compressive strength of the cement paste. Therefore, the performance of CH-based fly ash composites produced by low-temperature curing was superior to that of composites produced by high-temperature curing.

## 1. Introduction and Background

### 1.1. Research Background

Coal-fired power generation has become the first choice for filling the power generation gap due to its low price and high stability. Fly ash is a by-product of coal combustion in power plants, and the current annual production of coal ash worldwide is estimated to be around 600 million tons, with fly ash constituting about 500 million tons, or 75–80% of the total ash produced [1,2]. The well-considered use of fly ash generated from coal-fired power generation will result in environmental and economic benefits. For example, the total amount of coal ash produced by Japan’s power industry and general industry exceeds 10 million tons, of which 97.4% is used and 96.3% is used, respectively, as a raw material for the cement industry. In general, concrete containing fly ash as an additive or as a replacement for cement has improved durability [3]. The higher compressive strength of concrete containing fly ash is associated with improved bonding between aggregates and slurry and with a denser microstructure owing to an altered pore size distribution [4]. Pumice and perlite used as fine aggregates have been experimentally shown to have good freeze–thaw resistance in their concrete mixtures which can better improve the durability of concrete [5]. Dong Lu et al. also significantly reduced the cost of concrete by using conductive aggregates to enhance the durability of concrete [6]. As an important method to evaluate the strength distribution of materials, the Monte Carlo simulation (MCS) method can further analyze the statistical correlations at macroscopic and mesoscopic scales [7]. Currently, in addition to replacing cement, fly ash is also used in large quantities in other industries, greatly increasing its waste reuse. More and more scholars are using fly ash in the preparation of geopolymers and related performance studies. Rui Xiao et al. [8] analyzed and predicted various properties of alkali-activated materials (AAM) by establishing thermodynamic modeling. The effectiveness of fly ash for AAMs and the accuracy of the prediction of various indices of alkali activation using thermodynamic modeling were confirmed.

The pozzolanic reaction is an important factor affecting the dynamic properties of concrete, and thus considerable effort has been made over the decades to study the pozzolanic reaction of fly ash. Narmluk et al. studied the effect of fly ash on the hydration kinetics of Portland cement in fly ash–Portland cement mixtures with low water/binder ratios at three curing temperatures [9,10]. Wang et al. [11] proposed a multiphase model to simulate the pozzolanic reaction of low-calcium fly ash and slag. Zhang et al. [12] studied the effects of non-evaporative water content on the hydration of high-volume fly ash cement pastes in a large number of fly ash cement pastes cured at room temperature. Termkhajornkit et al. showed that the strong dependence of fly ash–cement concrete on curing conditions did not come from the hydration degree of the fly ash, but from the hydration degree of the cement, especially barite [13]. However, the removal of carbon from fly ash is also an important factor affecting its kinetic characteristics and other properties.

### 1.2. Flotation Method

In recent years, the decarbonization of fly ash has been a focus of research. Current common methods of reducing the unburned carbon in fly ash are dry separation (e.g., screening and friction electrostatic separation) and wet separation (e.g., foam flotation and gravity separation) [14,15]. Sieving exploits the size distribution characteristics of unburned carbon to achieve removal, as demonstrated by the measurable reduction of unburned carbon only in fine particles [16]. The research on electrostatic separation is relatively mature [17,18,19,20]. Triboelectrostatic separation of fly ash is usually accomplished by triboelectrification of the fly ash followed by passing it between two oppositely charged electrodes [21]. The separation relies on the enrichment of unburned carbon at a fine size fraction [16]. At present, this method can successfully remove unburned carbon in fly ash [22,23]. Soong et al. combined sieving and triboelectrostatic separation and obtained final products with a low loss on ignition (LOI) of 2.5% and a high LOI of 60% [24,25,26,27]. The main wet separation method of separating unburned carbon from coal ash is the foam flotation method, in which the dispersion of an oily collector is increased by emulsifying the collector with a synergist [28]. The emulsification increases the adhesion of the collector to the coal fly ash, thereby improving the flotation kinetics [24]. With the appropriate amount of collector and foaming agent, ash products can be obtained after the fourth flotation with an LOI of 2.13% and a recovery rate of 94.21%; however, the cost of flotation reagents is relatively high [25,29]. In contrast to froth flotation, gravity separation takes advantage of the density differences between unburned carbon particles (~1.8 g/cm^3^) and ash particles (~2.5 g/cm^3^) [30,31]. Gravity concentration, which is one of the oldest separation techniques, relies on density differences between minerals to facilitate separation. For fine particle separation, centrifugal acceleration is applied to the particles to increase the settling rate and enhance the separation [32]. By studying previous separation methods, we have proposed a new flotation method for improving fly ash quality and lowering environmental load. Our method can improve fly ash that does not meet the Japanese Industrial Standards (JIS) to a quality comparable to the JIS, and we have developed our own equipment for the method [33].

### 1.3. Research Purpose and Significance

Differences in curing temperatures may affect the pozzolanic reaction of concrete containing fly ash instead of cement. The strength properties and hydration reactivity of modified fly ash can be clarified by focusing on the reaction between calcium hydroxide (CH) and silica components leached from zeolitic materials. The CH and chemically bound water contents are the crucial parameters in the pozzolanic reaction mechanism of fly ash. Wang et al. [34] determined the CH content by using a modified tangential method, which considered the effects of both the decomposition of the C-S-H gel and the carbonation that may occur during sample preparation. They performed an experimental investigation of the pozzolanic reaction and the curing temperature dependance of low-calcium fly ash in cement systems, as well as the Ca-Si-Al ratios of fly ash-blended cement paste [34]. Because of the importance of the pozzolanic reaction in fly ash cement system, more and more scientists and scholars are currently investigating the promotion of secondary volcanic ash reactions. Finely ground glass powder promotes the reactivity of secondary volcanic ash [35]. In addition, the glass powder has the effect of controlling the alkali–silica reaction by indirectly reducing silica [36]. Concrete made of fly ash instead of cement generally has better properties than conventional concrete due to the pozzolanic reaction of fly ash. However, the physical and chemical properties and variability of fly ash also affect its engineering applications. For example, Jatuphon Tangpagasit et al. [37] studied the mortar and volcanic ash reaction using fly ash from Thailand with 20% fly ash instead of cement. The experimental results show that the compressive strength and activity vary even if the origin is the same and depend on the particle size of the fly ash. However, the pore size distribution of cement slurry made with high-calcium fly ash produced by the same coal-fired power plant had different effects on other properties due to the variation in the chemical properties of the fly ash [38]. Therefore, in the present study, we used two types of fly ash from different carbon sources and one type of fly ash conforming to the JIS.

The compressive strength development of cement paste subjected to a history of curing temperatures that mimic the internal temperature of concrete in practice has been investigated, but the compressive strength properties of modified fly ash, its pozzolanic reactivity, and the resulting hydrate are unclear. Although considerable efforts have been made to study pozzolanic reactions, most studies have focused on investigating specific temperatures (e.g., early high-temperature curing or room-temperature curing) and single sources of fly ash. This information is insufficient for understanding the properties of fly ash and its pozzolanic reactions fundamentally. In this work, we investigated the effect of curing temperature on the compressive strength of cement pastes with different coal sources of modified fly ash as a cement admixture, and the effect of curing temperature on the pozzolanic reaction of CH paste.

## 2. Materials and Methods

### 2.1. Modification of Fly Ash

Fly ash quality is classified into four types from Class I to Class IV in JIS A 6201 “Fly ash for concrete”, revised in February 1999. Table 1 shows the quality of fly ash specified in JIS A 6201 [39]. In addition, the Architectural Institute of Japan, JASS 5 M-401: 2007 “As a binder”, describes the quality of fly ash when it is used as a binder in “Mixing design of concrete using fly ash, construction guidelines and explanations”, and it is specified in “Quality standard of fly ash used”. Therefore, if fly ash is used as the binder, the quality should meet the specifications of JASS 5 M-401:2007, and if it is not used as the binder, it should meet JIS Class II or IV standards. In this paper, a new flotation method was used [33]. Figure 1 shows the flotation equipment that we used. By adding kerosene (2%) as an unburned carbon trapping agent and pine oil (0.2%) as a foaming agent, fly ash containing a large amount of unburned carbon generated a large number of bubbles, and the modified fly ash from which unburned carbon was removed was prepared. The LOI of fly ash obtained by the new flotation method conformed to the JIS specifications.

### 2.2. Materials

To examine the pozzolanic reaction of fly ash, fly ash and CH were used to make the CH paste for samples other than those for compressive strength measurements. Cement paste was used for compressive strength measurements. Table 2 shows the materials used. Figure 2 shows JIS fly ash and fly ash under a microscope. Two types of corporate power generation ash emitted by private thermal power generation were used as the fly ash admixture. Fly ash P was modified to conform to JIS Class IV (PMFA ash) standards, and fly ash R was modified to conform to JIS Class II (RMFA ash) standards.

Table 3 and Table 4 show the chemical and mineral compositions of the fly ashes as measured by X-ray fluorescence and the Rietveld refinement method, respectively. The effects on the chemical composition of the modification of fly ashes P and R were small. In addition, the proportions of mullite and quartz, which are the main minerals in fly ash, increased. The amount of the glass phase, which affects the pozzolanic reaction, was decreased by the modification, although the amount of glass phase was low in both the modified ashes and J ash. The glass phase amount in the modified R ash is less than half the previous value. Therefore, the modification of fly ash by the flotation method affected the quality.

### 2.3. Curing Method and Sample Preparation

As shown in Table 5. Samples of cement paste containing no fly ash (FA0) as the control group or 30% fly ash (FA30) were prepared. (In the following text: P-20 means the PMFA ash addition rate is 30% at a room temperature of 20 °C for curing.) For CH paste preparation, calcium hydroxide (CH) was used as a commercially available reagent (for chemistry), and a powder obtained by mixing equal amounts of FA and CH was used as a binder; the water/binder ratio (W/B) was 0.5.

The curing method used a sealed plastic column with a diameter of 50 × 100 mm. After mixing, the mixture was sealed in a cylindrical mold made of silicon. The sample was crushed at a predetermined age, the hydration reaction was stopped by immersion in acetone, D-drying was performed for 72 h using a vacuum dryer, the sample was crushed using an automatic mortar, and the powder was sieved to a particle size of 300 µm using a 300 µm sieve. The ages of the measurement samples were 3, 7, 28, and 91 days. The samples were cured under various curing temperature histories (Figure 3). Figure 4 shows the thermostat used to cure the sample. The temperature conditions were controlled by using a programmable constant-temperature bath. The conditions were constant curing at 20 °C, or they simulated the temperature in a concrete column during a standard period (pattern A), summer (pattern B), or winter (pattern C) during construction. The maximum initial temperatures for patterns A, B, and C were 50, 80, and 40 °C, respectively.

The samples are named by the fly ash type followed by the curing conditions. For example, P-20 denotes a sample containing PMFA ash that was cured at 20 °C, R-A denotes a sample containing RMFA ash that was cured with pattern A, and J-C denotes a sample containing JIS Class II ash that was cured with pattern C.

### 2.4. Measurement Method

The measurements included the chemical and mineral compositions of the fly ash, the compressive strength of the cement paste, the CH content of the CH paste, the fly ash reaction rate, and quantification of the hydrates. Compressive strength was measured according to JIS A 1108 “Concrete Compressive Strength Test Method”. The test system was made after all the materials were put into a special mixer; first was low-speed mixing for 30 s, followed by high speed mixing for 3 min and 30 s, and then the materials were sealed using a φ50 × 100 mm cylindrical plastic frame. After grinding the samples using an automatic grinder, we screened the powder samples below 300 microns for thermal analysis and reaction rate testing. Testing of calcium hydroxide content was conducted using a thermal analysis device. The weight loss values were calculated from room temperature to near 450 °C. This was to exclude the variation of water binding with the age of the material. The age of the samples was measured at 7, 28 and 91 days. The following Equation (1) was used to calculate the amount of calcium hydroxide.
The amount of Ca(OH)_2_ = W_s_ × 74/18/m_s_(1)
where:

W_s_ is the weight loss value around 450 °C;

m_s_ is the content of sample taken (mg).

The fly ash reaction rate was measured by using the centrifugation method from the selective dissolution method reported by Asuka et al. [40] and the calculation method from the study of the Kobayakawa River [41]. The sample was centrifuged (H-40F, KOKUSAN), and the insoluble residual amount was calculated according to JIS R 5202. The formula for calculating the reaction rate is shown in Equations (2) and (3).
(2)$$b_{d}=(f_{r}\times f_{i}-a_{d})/(f_{r}\times f_{i}/100)$$
(3)$$a_{d}={a_{d}}^{\prime }/(1-IG_{d}/100)$$
where:

b*_d_* is the reaction ratio;

*a_d_* is the insoluble residual volume collected by bound water at d days;

*a_d_′* is the insoluble residual volume collected;

*f_r_* is the fly ash addition rate;

*f_i_* is the insoluble residue of unhydrated fly ash;

*IG_d_* is the loss on ignition of the CH paste.

The CH paste was analyzed by powder X-ray diffraction (Smart Lab, Rigaku) and the Rietveld refinement method with PDXL described by Ri et al. [42].

For CH paste quantification, α-Al2O3 (10 mass%) was used as the internal standard substance. Alite (C3S), belite (C2S), interstitial (C3A, C4AF), periclase, dihydrate gypsum (gypsum), ettringite, various minerals of CH, hydrate, calcium silicate hydrate (C-S-H), and noncrystalline solids were contained in FA. The age of the measured material was 3, 7, 28, or 91 days.

## 3. Results and Discussion

### 3.1. Properties of Fly Ash

Figure 5 shows the particle size distributions of different types of fly ash. There were clear differences in the particle size distribution of fly ash before and after modification. Particle size has a strong effect on the flotation behavior of the mineral particles [43,44]. The collision efficiencies of the particles and bubbles and the particles and collector for fine particles are lower than that for an optimal flotation particle size [45]. Intermediate-sized particles have a high flotation rate, resulting in more efficient flotation [46]. Although we focused on the effect of temperature history on the reaction rate and compressive strength of cured products, particle size distribution is also a main factor affecting flotation and various properties. In the hydration process, the smaller the particle size, the faster the dissolution rate and the higher the hydration degree [47]. The particle size distribution was only part of the analysis of fly ash, and its important effects will be verified in future studies.

Figure 6 has two parts. One part is the activity indices of different unmodified fly ashes. As shown in the figure, the activities of the three different fly ashes vary greatly. The activity indices of PFA and RFA change with the increase of time. However, PFA conforms to the JIS II fly ash standard at 28 and 91 days before modification.

Figure 6a shows that the activity indices of the unmodified fly ashes P and R varied greatly over time. However, fly ash P conformed to the JIS Class II fly ash standard at 28 and 91 days before modification.

Figure 6b shows the activity indices of the modified fly ashes. The modified fly ashes had similar heat losses. The LOI of PMFA and RMFA was about 1.4%. The results showed that fly ash P, which had a good activity index before modification, failed to meet the standard 28 days after modification. At 28 days, only the RMFA activity index reached the standard of JIS Class II fly ash. However, at 91 days, the activity indexes of both modified fly ashes reached the standard. The results showed that the modified fly ash could replace cement because of its relatively stable performance. There are many factors affecting fly ash activity, including volume, density, porosity, specific surface area, and grain size of CaO minerals, that affect the later pozzolanic reaction [48,49].

### 3.2. Relationship between Curing Temperature and FA Reaction Rate

Figure 7 shows the fly ash reaction rates for samples with different curing temperature histories. The reaction rate generally increased from 3 to 91 days. Comparing the temperature history, the JIS Class II ash (J samples) showed the lowest reaction rates, except for pattern B. In addition, the reaction rates for samples P and R at a constant temperature of 20 °C were almost the same as for pattern A. The differences in solidification at a specific temperature among modified fly ashes from different coal sources were small.

Furthermore, modified ashes tended to exhibit higher reaction rates than JIS Class II ash in the order of pattern B > pattern A > pattern C > 20 °C at ages of 3 and 7 days. However, at 28 days, all ash values for pattern C were below that of ash cured at 20 °C, with differences ranging from 3.8% to 8.2% at 91 days. Thus, the reaction rate increased with higher temperature history for a given temperature. As with pattern C, when initially heated to 40 °C and cured at a temperature below 20 °C after 6 days, initial curing at low temperatures, with or without heat, is thought to slow down the pozzolanic reaction.

Fly ash is chemically involved in cement hydration in two ways [9]. First, cement produces Ca(OH)_2_ during hydration, and fly ash consumes Ca(OH)_2_ in the pozzolanic reaction. Second, both cement and fly ash consume water in their hydration reactions [50]. Otsuka et al. [51] found that in cement pastes containing 20% fly ashes with different physicochemical properties, the amount of fly ash glass and the grain specific surface area were major factors that determined the fly ash reaction rate. The grain specific surface area was in the order PMFA ash ≒ RMFA ash > JIS Class II ash. The amount of fly ash glass did not depend on the carbon source (Table 4). Therefore, the fly ash reaction rate depended more on the specific surface area than on the amount of fly ash glass.

The curing temperature is an important factor affecting the performance of fly ash concrete [52]. High-temperature curing promotes the pozzolanic reaction of fly ash [53]. Sato et al. [54] found that the pozzolanic reaction of fly ash occurred readily between 60 and 80 °C. Our study showed a similar trend, but the increase in the reaction rate for pattern B, which had the highest initial temperature of 80 °C, decreased after 7 days. A rapid hydration reaction at high temperature can cause the uneven distribution of hydration products, which may affect the change in porosity and reduce the compressive strength [55,56].

### 3.3. Relationship between Temperature History and CH Abundance

Figure 8 shows the change in the amount of CH over time for different temperature histories. The results confirmed the progress of the pozzolanic reaction, with the amount of CH decreasing for all ash types and curing conditions. The CH amounts in the CH pastes of samples P and R for all temperature histories and ages were lower than those for samples J and were similar. This is mainly because the calcium oxide content in the chemical composition of JIS Class II ash was higher than that in the PMFA and RMFA ashes. When compared by temperature history pattern, the amount of CH in all fly ashes decreased in descending order of temperature history for a given temperature. At 0 days, pattern C had the smallest amount of CH, whereas at 91 days the amount of CH was smallest for ash cured at 20 °C. Figure 9 shows the relationship between the fly ash reaction rate and the amount of CH. There was a correlation between the fly ash reaction rate and the CH amount for all ash types and curing conditions. Irrespective of the carbon source and curing temperature of the fly ash, the CH amount and the reaction rate were always negatively correlated, and the correlation was high. These results showed that the reduction in CH depended on the curing temperature and the fly ash reaction rate. The general trend was that samples that consumed a lot of CH also had higher fly ash reaction rates.

In the pozzolanic reaction, fly ash consumes CH and generates hydrates, such as C-S-H. Saensoy found that fly ash cement paste was better hydrated than ordinary Portland cement [57]. In addition, Wang reported that the reaction rate of fly ash is not only related to temperature [34] but may also be correlated with the content of mullite. This result was confirmed by further studies on mullite. The experimental results show that the reactivity of fly ash and its insoluble content are affected by the content of crystalline mullite [58]. Meguro et al. [59] determined the amount of CH consumed by the pozzolanic reaction based on the reaction rate of fly ash regardless of the effect of the properties of fly ash. In the present study, similar results were obtained for each type of fly ash. Therefore, the amount of CH consumed by the pozzolanic reaction was related to the reaction rate of fly ash even for curing with a temperature history.

### 3.4. Relationship between Temperature History and C-S-H Production Amount

Figure 10 shows the amount of C-S-H produced for each temperature history. The amorphous content was calculated experimentally. Similar to Hoshino et al. [42], we calculated from quantitative results for α-Al2O3 and assumed that most amorphous materials are C-S-H.

From 3 to 91 days, there was a general increase in C-S-H production for patterns A and C and at a constant temperature of 20 °C, whereas pattern B showed no increase (Figure 10). Thus, the increase in the production of C-S-H stagnated from 3 days for high-temperature curing. At 3 days, production was the largest for P samples in all temperature histories. Even at 91 days, the C-S-H production for P samples for curing at 20 °C was the maximum observed for pattern A and B. Sample P-A showed the largest production at 91 days, and the production in the other P samples was in the order P-20 > P-C > P-B. Sample J-C reached its maximum at 91 days, and there was no significant difference except for J-C. Furthermore, at 91 days, R samples converged to about 40% for all temperature histories. According to the differences in the characteristics of fly ash, it can be divided into types that are easily affected by the temperature history and types that are not affected by the temperature history. However, for initial high-temperature curing, fly ash from different coal sources has little effect on C-S-H generation. For the JIS Class II ash, the differences in C-S-H generation were small among the four patterns. After flotation modification, the properties of RMFA were more stable than those of PMFA.

Compared with the reference sample, the C-S-H of the fly ash blended with Portland cement was characterized by lower Ca/Si and higher Al/Si ratios [60]. The coal sources of PMFA and RMFA were different; thus, their proportions of aluminum and silicon were also different. The change in the composition of C-S-H is related to the silicon content provided by fly ash upon dissolution [61]. Together with the Ca/Si ratio and silicate structure, Si–OH and Ca–OH groups are important in determining the nanostructure of C-S-H [62]. The reactivity of these groups increases with temperature, which changes the amount of C-S-H.

### 3.5. Relationship between Temperature History and Compressive Strength

Figure 11 shows the compressive strength of all cement paste samples cured with the four temperature histories. O samples are the control cement paste samples (FA0), which contained no fly ash. The compressive strengths of P, R, and J samples were lower than those of O samples under all curing conditions. At 3 days, the strengths of R and J samples decreased with the decrease of curing temperature. The increase in strength was largest at 7 days for a constant temperature of 20 °C. Sato et al. [63] observed the same trend in strength for curing at 20 °C when the temperature of the water-cured concrete reached 60 °C from the initial age to age 56 days. In the present work, comparing pattern A with curing at 20 °C, the O, P, and J samples showed the same trend in compressive strength at 28 days. The results suggested that although PMFA and RMFA both used the same method for flotation, they exhibited differences in chemical composition, physical properties, and chemical properties due to the different coal sources. The compressive strengths of the cement slurries containing modified fly ash and benchmark JIS Class II ash had large differences at the same curing temperature. Therefore, in addition to the curing temperature, the properties and sources of fly ash itself are strongly related to the compressive strength of the cement paste.

The structure of capillary pores becomes coarser in samples cured at higher temperatures [63], and the pore structure is changed by differences in temperature history. The strengths at 3 days were higher for pattern C than for curing at 20 °C except for P samples, but after 7 days, the strengths were lower for pattern C than for curing at 20 °C for all fly ash samples. However, because the strength increases did not stall even after 28 days, it is possible that compressive strength continues to increase in the long term. The compressive strengths after 28 days decreased in the order of P > R > J samples for curing at 20 °C and pattern A. However, for pattern B, after three days, the strengths for P samples were larger than those for R samples. In other words, when the strength development of modified fly ash samples occurred at 20 °C or with initial aging at 50 °C, they showed higher strength development than JIS Class II fly ash samples and showed higher strength development from high-temperature primary aging at 80 ℃; this experiment found that the strength development after 3 days was less than that of JIS Class II fly ash. These differences were caused by the hydration and reaction of modified fly ash and JIS Class II ash with different chemical compositions at different temperatures. Temperature affects the dissolution and precipitation reactions of calcium carbonate (CaCO3). Usually, a variety of crystalline polymorphs can be formed, such as calcite and aragonite [64]. It is also possible to form other allotropes, of which calcite forms a thermodynamically stable crystalline phase at ambient temperature and standard pressure [65]. These reactions explain the effect of temperature on the hydration reaction and strength.

### 3.6. Relationship between Compressive Strength and FA Reaction Rate

Figure 12 shows the relationship between the compressive strength of cement paste and the reaction rate of CH paste. A strong linear relationship was observed for a constant temperature of 20 °C, and a positive correlation was observed for pattern A. However, patterns B and C showed no correlation due to the different ash types. In other words, if the curing temperature changes, the development of strength cannot be explained only by the pozzolanic reaction. Therefore, the effects of the curing temperature on the cement hydration reaction and the pozzolanic reaction are different, and the compressive strength under high-temperature and low-temperature curing conditions has little effect on the cement hydration reaction and pozzolanic reaction.

The compressive strength of cement paste could be evaluated by the fly ash reaction rate without considering the effect of the type of fly ash for curing at 20 °C and an initial temperature of 50 °C (pattern A). In other studies, the strength development of mortars containing fly ash was attributed to the pozzolanic reactivity and the effect of fly ash [66,67]. The strength of hardened cement arises from physical bonds in addition to chemical bonds [68]; thus, in future work, the effects of the addition amount and particle size distribution of the fly ash, the specific surface area of the hardened cement, and the C/S ratio should be investigated.

## 4. Conclusions

In this experiment, two types of fly ash with no coal source were modified by flotation. The difference in particle size between RFA and RMFA was large, but the overall activity of the modified RMFA was better than that of PMFA. The effect of curing temperature on the pozzolanic reactions of different types of fly ash–cement mixtures depends on the temperature history of the whole process. The progress of the pozzolanic reaction was confirmed by the gradual decrease in the amount of CH for all fly ash types and curing conditions. Both P20 and R20 with a curing temperature history of 20 °C had higher compressive strengths in the later stages of the reaction than the samples with other curing histories. The specific conclusions are as follows:In this work, we found that flotation modification had little effect on the chemical composition of fly ash. The factor that affected the pozzolanic reaction was the lower glass content in fly ash before modification and the fact that the quality of the fly ash is affected by the use of flotation to modify the fly ash. RMFA has a particle size distribution in the range of 0.1–1 µm, which is about 4 times larger than that of the original ash, with a larger increase in small particle size than WMFA. WMFA is less active than RMFA in the first, middle, and late stages, but it can reach 96.53% at 91 days and can meet the JIS Class II ash specification.The reaction rate of fly ash did not depend on the amount of glass, and the reaction rates of the modified fly ashes for all periods were higher than that of JIS Class II fly ash. The highest early reaction rate was observed at an initial temperature of 80 °C, but the reaction rate increased slowly after 28 days. The reaction rate increases slowly after 7 days when the temperature is lower than 20 °C, and when the curing temperature is lower than 10 °C at 14 days, the average increase in response rate in the middle and late stages of the reaction is 5.55%. When the curing temperature is raised from the age of the feed and the temperature is continuously maintained at a high level, the increase in the reaction rate slows down after 28 days of feeding. In addition, curing at temperatures below 20 °C also slowed the increase in reaction rate.For all fly ashes, the amount of CH decreased with increasing curing temperature. Furthermore, the fly ash reaction rate was correlated with the amount of CH for all fly ash types and curing conditions. The amount of CH consumed by the pozzolanic reaction was related to the fly ash reaction rate during curing with a given temperature history. Initial high-temperature curing slowed down the production of C-S-H. The fly ash could be divided into types that were and were not affected by temperature history.The cement paste without fly ash developed high compressive strength under all curing conditions, and its compressive strength can reach up to 65.83 N/mm^2^. The compressive strength at three days increased with the initial curing temperature. After 7 days, the increase in strength was largest for curing at 20 °C. The modified fly ash samples exhibited higher strength development than JIS Class II fly ash cured at 20 °C and at initial temperatures of 50 and 80 °C. The strengths of modified fly ash samples were lower for curing at an ultra-high temperature.The relationship between the compressive strength of the cement paste and the fly ash reaction rate of the CH paste showed that the compressive strength of the cement paste was determined by the fly ash reaction rate, not by the fly ash properties.

## Figures and Tables

**Figure 1 materials-16-02645-f001:**
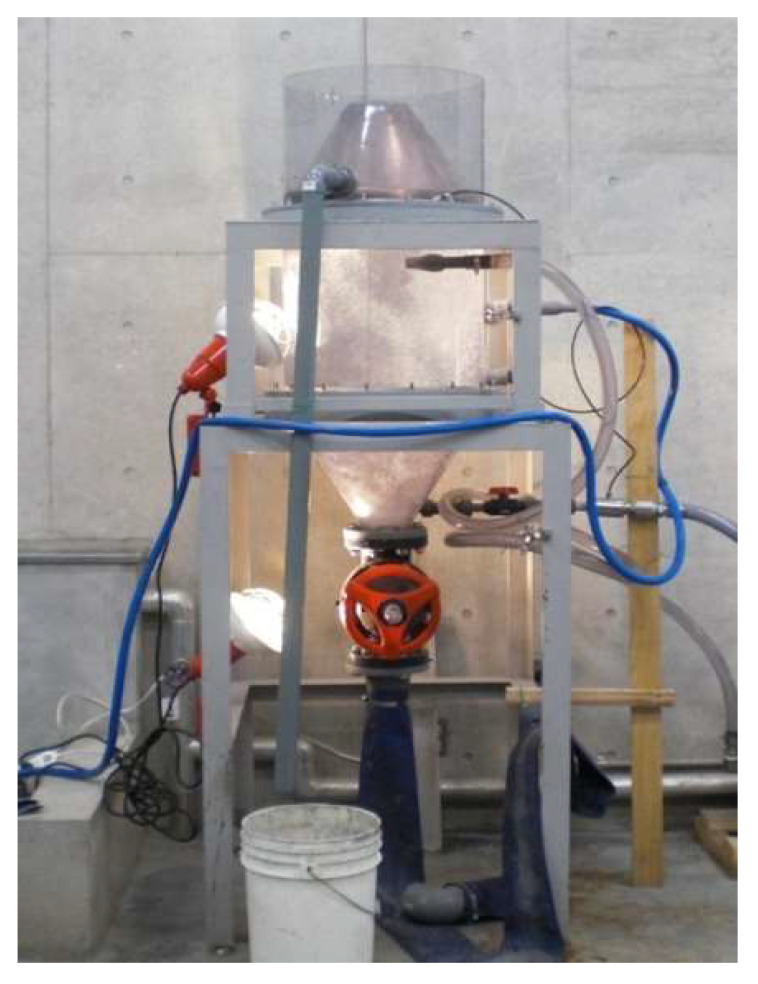
The flotation equipment for fly ash.

**Figure 2 materials-16-02645-f002:**
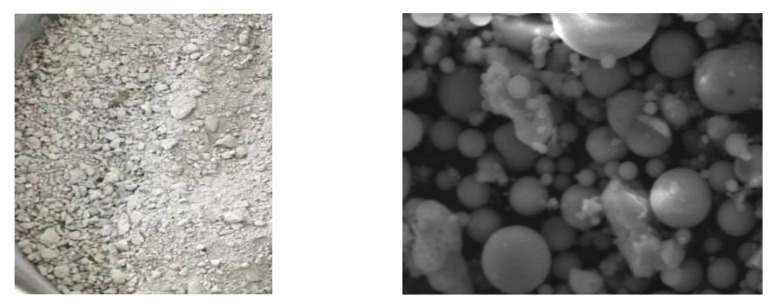
Fly ash (**left**) and electron micrograph (**right**).

**Figure 3 materials-16-02645-f003:**
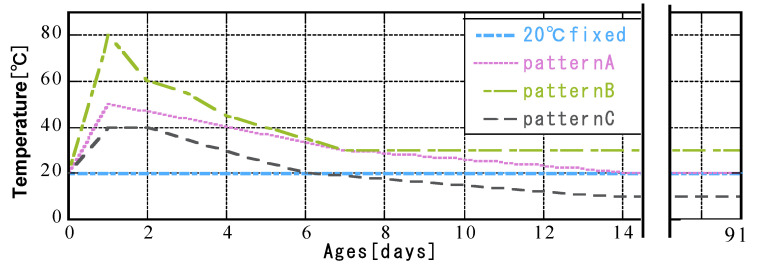
Curing temperature history.

**Figure 4 materials-16-02645-f004:**
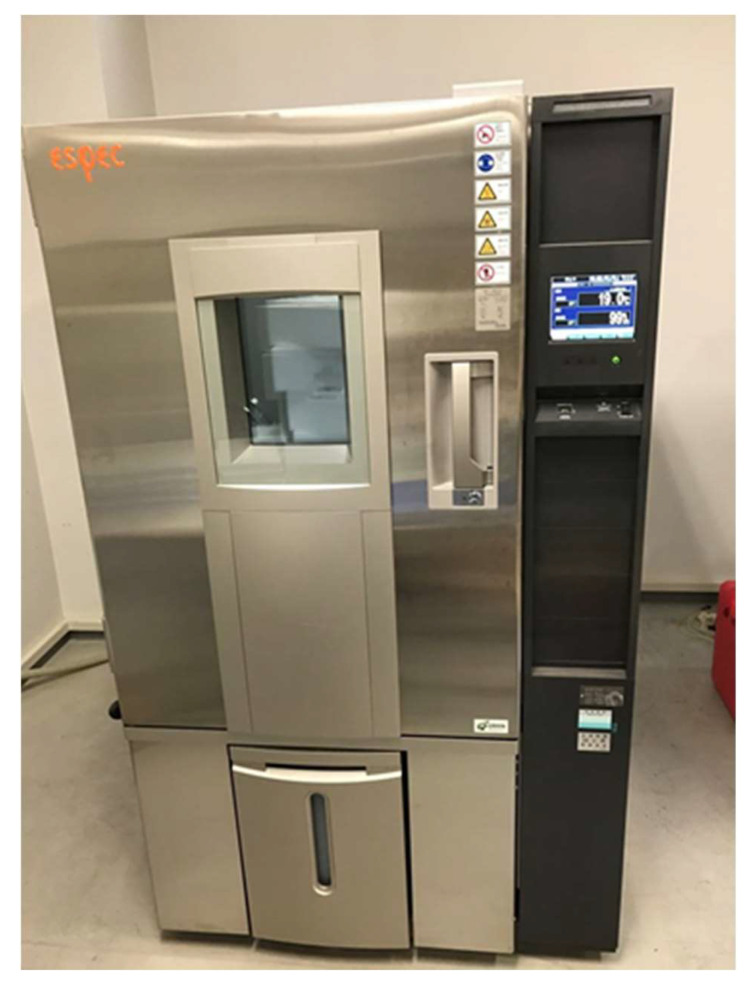
Thermostat machine for sample curing.

**Figure 5 materials-16-02645-f005:**
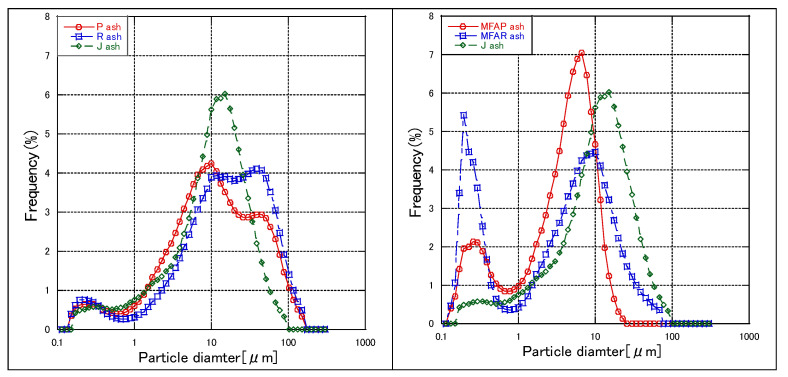
The particle size distribution of fly ash.

**Figure 6 materials-16-02645-f006:**
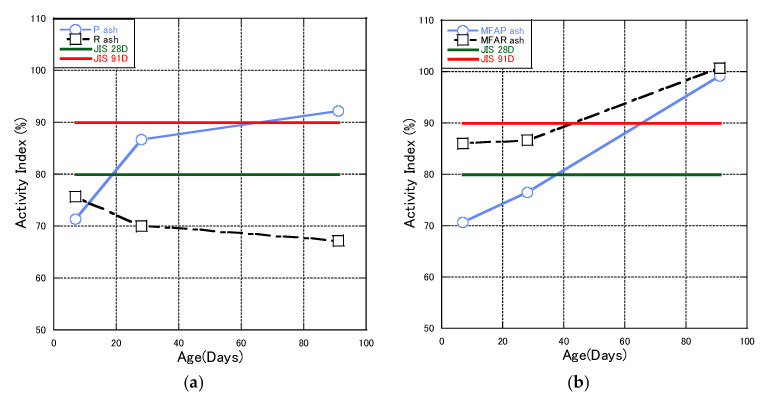
The activity indices of fly ashes.

**Figure 7 materials-16-02645-f007:**
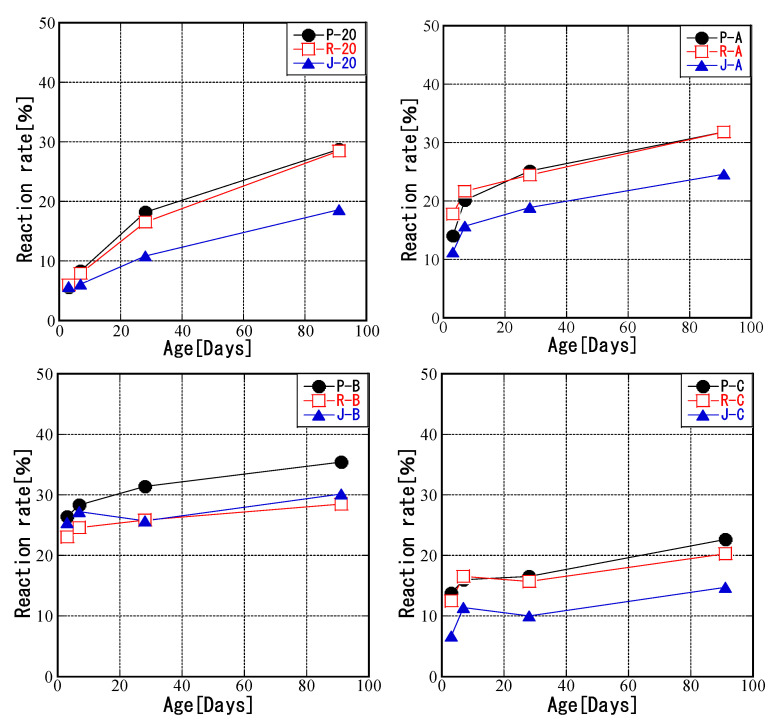
FA reaction rates.

**Figure 8 materials-16-02645-f008:**
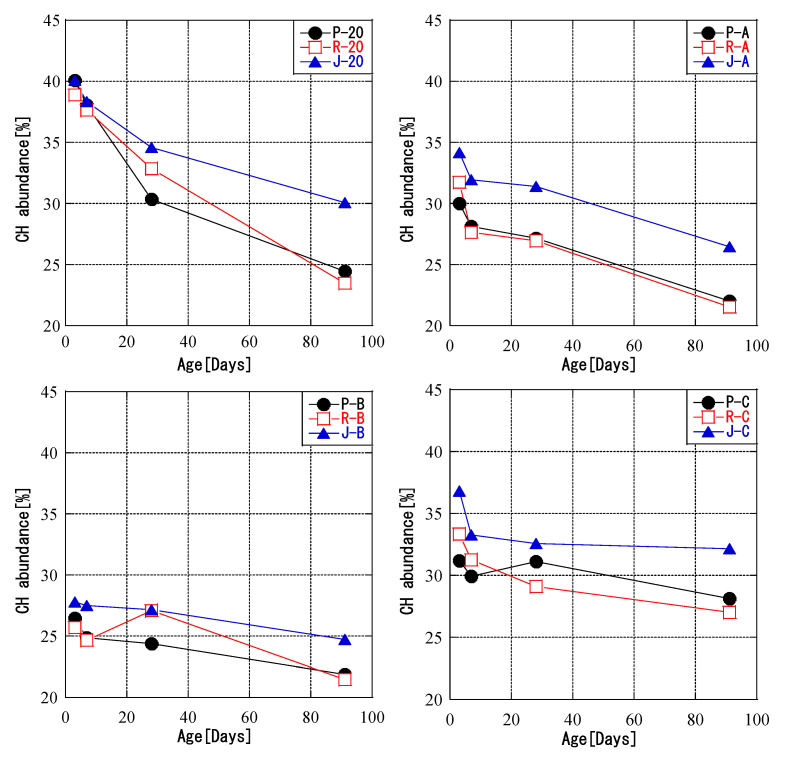
CH abundance.

**Figure 9 materials-16-02645-f009:**
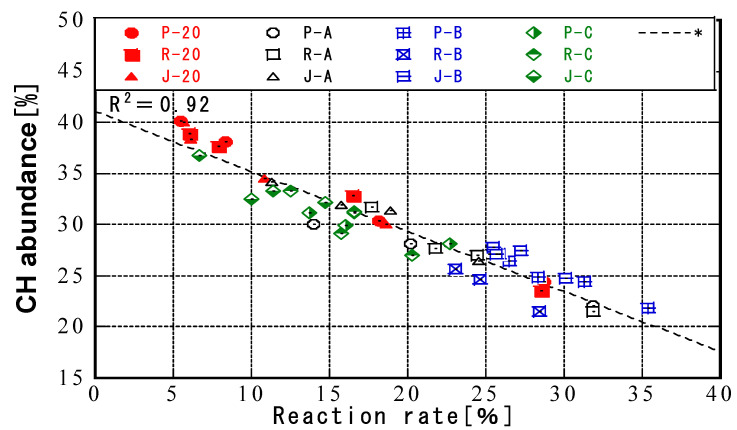
Relationship between FA reaction rate and CH abundance.

**Figure 10 materials-16-02645-f010:**
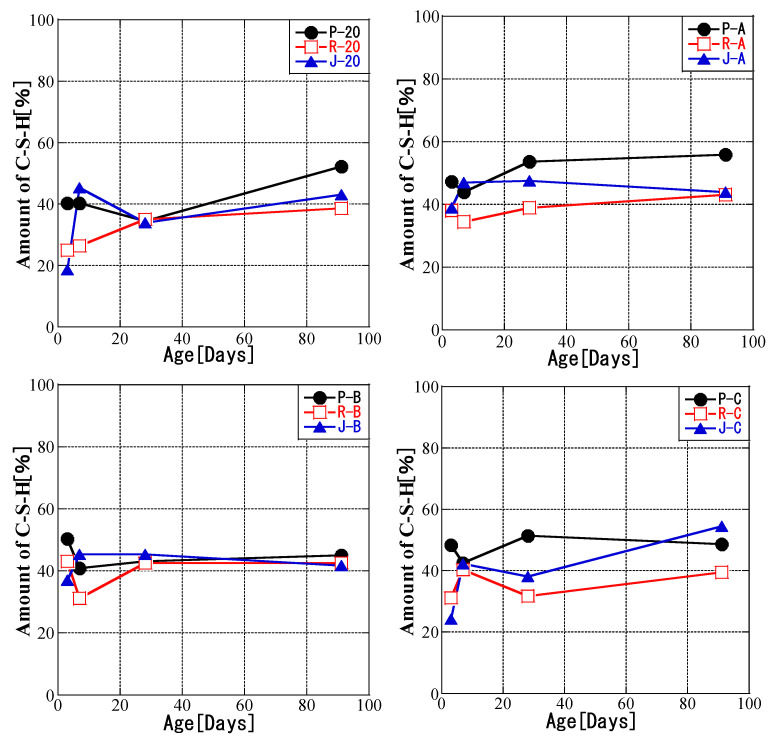
C-S-H production amount.

**Figure 11 materials-16-02645-f011:**
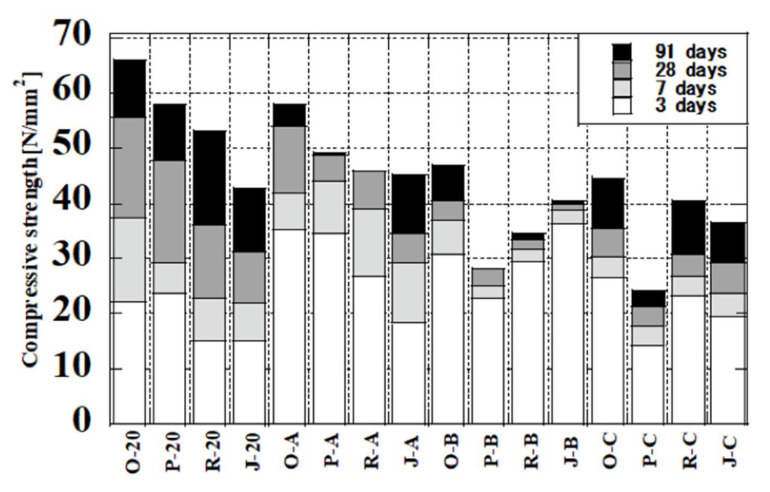
Compressive strength.

**Figure 12 materials-16-02645-f012:**
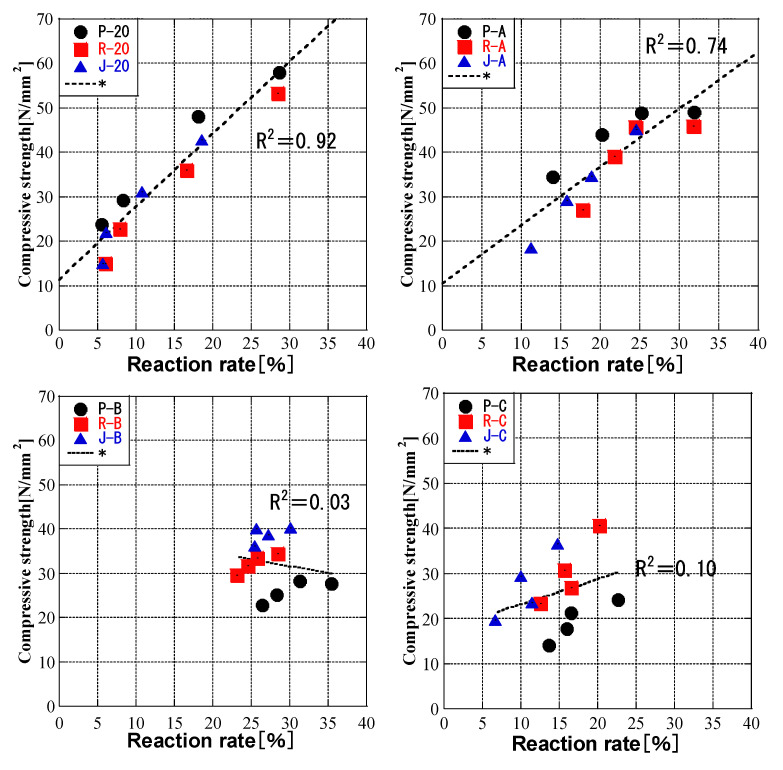
Relationship between compressive strength and FA reaction rate.

**Table 1 materials-16-02645-t001:** The Japanese Industrial Standards for fly ash for concrete.

Type		JIS Specification Year 1958	JIS Specification Year 1974	JIS Specification Year 1966
loss on ignition (%)		≤5	≤5	≤5.0
Power	45 m sieve residue (Tuna sieving method: %)	≤25	__	≤40
	Specific surface area (cm^2^/g) Blaine method	≥2700	≥2400	≥2400
Unit water ratio (%)		≤100	≤102	
Flow ratio (%)				≥92
Compressive strength ratio (%)	28D	≥63	≥60	
Activity index(%)	28D			≥80
	91D			≥90
Density (g/cm^3^)		≥1.95	≥1.95	≥1.95
Humidity (%)		≤1	≤1	≤1.0

**Table 2 materials-16-02645-t002:** Materials.

Symbol		Type	Physical Characteristics
Cement	C	Portland cement	Density: 3.16 g/cm^3^
Admixture	FA	Raw ash fly ash (P ash)	Density: 2.30 g/cm^3^,
			Specific surface area: 5070 g/cm^3^
			Loss on ignition: 12.9%,
			Flow value ratio: 96.7%,
			28 days, 91 days Activity index: 86.7%, 92.2%
		Modified fly ash equivalent to JIS type IV (PMFA ash)	Density: 2.41 g/cm^3^,
			Specific surface area: 5520 g/cm^3^
			Loss on ignition: 1.7%,
			Flow value ratio: 111.1%,
			28 days, 91 days Activity index: 76.5%, 99.2%
		Raw ash fly ash (R ash)	Density: 2.19 g/cm^3^,
			Specific surface area: 4200 g/cm^3^
			Loss on ignition: 8.5%,
			Flow value ratio: 66.6%,
			28 days, 91 days Activity index: 70.0%, 67.2%
		Modified fly ash equivalent to JIS type II (RMFA ash)	Density: 2.41 g/cm^3^,
			Specific surface area: 5480 g/cm^3^
			Loss on ignition: 1.2%,
			Flow value ratio: 91.2%,
			28 days, 91 days Activity index: 86.8%, 100.7%
		Fly ash (J ash) JIS type II—certified product	Density: 2.30 g/cm^3^,
			Specific surface area: 4000 g/cm^3^
			Loss on ignition: 1.3%,
			Flow value ratio: 111%,
			28 days, 91 days Activity index: 90.0%, 102.0%
Water	W	Tap water	——

**Table 3 materials-16-02645-t003:** Chemical composition of fly ash.

FA	SiO_2_	Fe_2_O_3_	Al_2_O_3_	CaO	K_2_O	TiO_2_	SO_3_	MgO	ZrO_2_
P FA	44.70	22.80	16.00	5.33	3.72	2.13	2.12	1.84	0.71
P MFA	46.70	18.90	16.90	3.69	3.37	1.84	0.22	1.96	0.70
R FA	59.70	9.53	16.90	2.82	3.16	4.02	1.63	1.16	0.66
R MFA	62.40	8.70	17.60	2.30	3.06	3.49	0.14	1.14	0.73
J	54.80	10.20	13.00	13.70	1.83	2.97	1.03	1.20	0.77

**Table 4 materials-16-02645-t004:** Mineral composition of fly ash.

FA	Mullite (%)	Quartz (%)	Magnetite (%)	Glass (%)
P FA	13.26	9.68	2.70	54.20
P MFA	14.13	12.87	1.33	42.58
R FA	18.48	17.81	1.33	45.58
R MFA	20.32	21.29	0.77	37.74
J	9.86	8.26	0.97	59.13

**Table 5 materials-16-02645-t005:** Cement paste mix proportion.

Name	W/C (%)	Unit Weight (kg/m^3^)		
		W	C	FA
FA0	50.0	180	360	0
FA30	50.0	180	252	108

## Data Availability

The data presented in this study are available on request from the corresponding author. The data are not publicly available due to privacy restrictions.

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
