# Peer review of "The Effects of Curing Temperature on CH-Based Fly Ash Composites"

_materials, 2023, doi:10.3390/ma16072645_

Round 1
Reviewer 1 Report
Topic: I think the topic is too long
it could reframed as: ''The effects of curing temperature on CH-based fly ash composites''
Abstract:
The abstract is too sketchy and lacks clarity in terms of actual outcome of the research. Authors should clearly summarise the main findings numerically.
1. Introduction
This section should be headed as follows:
1. Introduction and Background
Authors should include sub-sections under this section, such 2.1, 2.2,.......
Conclusion
Authors must summarise their main findings
Reviewer 2 Report
This is a carefully done study and the findings are of considerable interest. A few minor revisions are list below.
1)The abstract is generally well written, however in terms of content it is generic, i.e., the authors lack an in-depth study of the quantitative results of this research.
2)The list of references is not in our style. It is close but not completely correct.
3)Some references are outdated; some are even very old. It is strongly recommended to use recent references.
4)“The higher compressive strength of concrete containing fly ash is associated with improved bonding between aggregates and slurry and with a denser microstructure owing to an altered pore size distribution.” is unclear. The effect of fines and aggregates is very important in mechanical properties of concrete, and it should be discussed. I suggest you consider including them in your revised manuscript:
Monte Carlo simulations of deformation behaviour of unbound granular materials based on a real aggregate library
Perlite and Rice Husk Ash Re-Use As Fine Aggregates in Lightweight Aggregate Structural Concrete—Durability Assessment
Carbon nanotube polymer nanocomposites coated aggregate enabled highly conductive concrete for structural health monitoring
5)“The amount of CH was measured by using thermogravimetric analysis,The tests were conducted in accordance with the Concrete Testing and Analysis Manual of the Japan Concrete Association.” need to be modified.
6)Please added some plots of samples and devices.
Reviewer 3 Report
This paper explored the effect of curing temperature on calcium hydroxide (CH)-based fly ash composites. Fly ashes from different carbon sources were used to make CH-based composites and the compressive strength, reaction rate, CH content, and C-S-H generation were analyzed. Overall speaking, this is a very interesting study. The experiments were well designed and the conclusions are well supported by the results. I only have some suggestions.
1. The first paragraph should have a brief introduction on the beneficial utilization of fly ash in cement and concrete industry. Fly ash can also be used to make alkali-activated materials (or sometimes called geopolymer) which have been considered as sustainable cements. Recently published papers should be reviewed. (e.g.,"Analytical investigation of phase assemblages of alkali-activated materials in CaO-SiO2-Al2O3 systems: The management of reaction products and designing of precursors. Materials & Design, 194, p.108975.")
2. Line 34: "The higher compressive strength of concrete containing fly ash is associated with improved bonding between aggregates and slurry". This statement needs proper citations.
3. The pozzolanic reaction which contributes to the formation of secondary C-S-H should be well described. There are also new pozzolans for cement industry such as waste glass powder.
4. Line 120: "By adding kerosene (2%) ". The authors should justify the treatment method. And how will the carbon in fly ash influence the properties of concrete?
5. The dimension of your samples and loading rate for mechanical property tests should be specified.
6. Generally, the authors should have "conclusions" as the final section of a research paper.
Round 2
Reviewer 3 Report
This paper has been revised based on the comments.